# Personal Autonomy as Quality of Life Predictor for Multiple Sclerosis Patients

**DOI:** 10.3390/jcm9051349

**Published:** 2020-05-05

**Authors:** Rodica Padureanu, Carmen Valeria Albu, Ionica Pirici, Radu Razvan Mititelu, Mihaela Simona Subtirelu, Razvan Aurelian Turcu-Stiolica, Harri Sintonen, Vlad Padureanu, Adina Turcu-Stiolica

**Affiliations:** 1Department of Biochemistry, University of Medicine and Pharmacy of Craiova, 200349 Craiova, Romania; zegheanurodica@yahoo.com (R.P.); razvanmititelu@rocketmail.com (R.R.M.); 2Department of Neurology, University of Medicine and Pharmacy of Craiova, 200349 Craiova, Romania; carmenvaleriaalbu@yahoo.com; 3Department of Anatomy, University of Medicine and Pharmacy of Craiova, 200349 Craiova, Romania; danapirici@yahoo.com; 4Department of Pharmacoeconomics, University of Medicine and Pharmacy of Craiova, 200349 Craiova, Romania; mihaela.subtirelu@yahoo.com (M.S.S.); adina.turcu@gmail.com (A.T.-S.); 5Trueman Consulting, 200528 Craiova, Romania; razvan.turcu@gmail.com; 6Department of Public Health, University of Helsinki, 00100 Helsinki, Finland; harri.sintonen@helsinki.fi; 7Department of Internal Medicine, University of Medicine and Pharmacy of Craiova, 200349 Craiova, Romania

**Keywords:** multiple sclerosis, quality of life, autonomy

## Abstract

Multiple sclerosis (MS) is a chronic, severe disease, characterized by a progressive alteration in neuronal transmission, which decreases personal independence and quality of life (QoL). This study aimed to investigate the relationship between QoL and personal autonomy in patients with MS, as well as its correlation with age, educational level, and diseases severity. Twenty-six MS patients were followed-up for six months. All patients completed the 15D questionnaire two times: at T0, when they started a new treatment, and at T1 after six months of treatment. At the end point, all patients completed the Personal Autonomy Questionnaire. The average patient age was 43 years (SD = 10), and 89% of them were female. The mean severity and duration of MS were 3.5 (SD = 1.75) and 9.5 (SD = 5.1), respectively. The average QoL of MS patients at T0 was 0.66 (SD = 0.18), and that at T1 was 0.71 (SD = 0.16). The scores of patients with different types of MS, i.e., relapsing–remitting MS (RRMS) or secondary progressive MS (SPMS), were compared. SPMS patients were older than RRMS patients (mean age 47.5 vs. 39.7 years; *p* = 0.032), and more RRMS patients were working (0.014). SPMS patients described the same QoL and personal autonomy as RRMS patients. Results from bivariate correlation analyses showed a significant relationship between QoL and age, education, and severity of MS. Also, the analysis showed no significant correlation between QoL and personal autonomy.

## 1. Introduction

Multiple sclerosis (MS) is a chronic, severe disease, affecting the central nervous system, often debilitating and leading to a progressive alteration of neuronal transmission [1] associated with a great variety of symptoms. The optimal therapeutic intervention and the etiology of the disease are far from complete understanding, even though a lot of research was made in the last decades [2,3,4]. On average, MS affects women two to three times more often than men. [5,6]. There are four forms of MS, the most common of which are relapsing–remitting MS (RRMS, 85%) and primary progressive MS (PPMS, 10%). Symptoms in MS are unpredictable and variable and include fatigue (80% of patients), abnormal sensations (paresthesias, pain, Lhermitte’s sign), motor symptoms (weakness, spasticity spasms), visual dysfunction, tremor, gait impairment, brainstem signs (diplopia, dysphagia, internuclear ophthalmoplegia), cognitive abnormalities, and depression symptoms [7,8]. Fatigue is associated with various health problems and is one of the most common signs of the disease. Fatigue can appear in all stages of the disease and can affect patients’ quality of life (QoL), emphasize anxiety and depression, and affect sleep patterns and the motor function [9,10,11]. In MS patients, because of cognitive impairments (lack of concentration, memory loss, deficits in information processing, and impaired reasoning skills), critical thinking and judgment ability are significantly affected. Another problem for these patients is to accept their condition, which greatly influences the desire to seek (additional) medical support and/or to adhere to chronic treatment plans [12]. If patients can satisfy their daily needs, managing the various symptoms of the disease and coping well with difficulties in everyday life, they will improve their QoL [13]. Studies have reported that fatigue reduces physical stamina, limits social duties, and interferes with performance of responsibilities at work and at home, substantially impacting the QoL of patients with MS [7]. Social relationships of patients with MS are limited as are their daily activities, because they have to take their medications (cortisone, interferon, or monoclonal antibody) [14]. In cross-sectional studies of patients with MS, fatigue has been inversely correlated with aspects of QoL [15,16] and is responsible for a poor QoL, even more than spasticity, weakness, and urinary or motor disorders. Patients with MS, compared to the general population, report having more severe depression and anxiety [17]. Several studies show that women with MS are significantly more anxious [18], and high levels of anxiety are associated with higher levels of disability [19]; also, low education is related to high levels of anxiety [20,21]. 

Health-related behavior in MS patients was assessed on the basis of patient autonomy regarding different tasks [22]. Personal autonomy (PA) consists in the ability to control one’s own life associated with the feeling that it is possibile to exercise this control and make an informed decision [23]. Personal autonomy is measured through four dimensions: cognitive autonomy, behavioral autonomy, emotional autonomy, and value autonomy. Each of these areas of autonomy is essential to determine patients’ QoL. It was shown that MS patients prefer more autonomous roles than patients with other health conditions [24] and that MS patients from Northern Europe might prefer more autonomous roles than those from Southern Europe [25].

Cognitive autonomy consists in the ability to reason independently and make decisions without overly seeking social validation, in a sense of trust in oneself, and in the belief of having the opportunity to choose. For a healthy person or a patient, true autonomy is not possible, given that he/she exists within relationships: people cannot make decisions without the pressure of other people involved in these relationships.

Steinberg et al. [26] defined emotional autonomy as independence from parents and equals. It refers also to emotions, personal feelings, and patients’ relationships with other people [27]. Emotional autonomy overlaps to a great extent with self-confidence [28].

Behavioral autonomy means making decisions independently and acting accordingly. Value autonomy is expressed in the creation of one’s own set of beliefs and principles, resistant to the pressure of others [27].

Self-reported QoL increases with the attainment of autonomy associated with social inclusion, depending on the perceived ability to maintain autonomy and control [29]. 

This study aimed to find potential correlations between QoL and personal autonomy of patients with MS. The QoL indicator is useful to assess beneficial changes in patients with MS after treatment. We also describe also the characteristics of secondary progressive MS (SPMS) and RRMS patients.

## 2. Materials and Methods

### 2.1. Ethical Issues

This research was approved by the Academic and Scientific Ethics and Deontology Committee of the University of Medicine and Pharmacy in Craiova (Registration No. 96/2019) according to European Union Guidelines (Declaration of Helsinki). All the patients signed an information and acceptance form to be included in the present study.

### 2.2. Participants

A number of 26 patients with MS were included in this study. The demographic variables were: age, gender, marital status, employment status, level of education, urban/rural environment. The participants were selected based on the following inclusion criteria: patients over 18 years old were divided in two groups, an RRMS group and an SPMS group, according to McDonald 2010 criteria for MS diagnostics. Exclusion criteria were: significant cognitive impairments, other clinical relevant systemic diseases, or other treatments that could affect the course of the study, restrict the patient ability to read and interpret the study information, adhere to the rule of the protocol of the study, or complete the study. The duration and the severity of the disease were considered in this study. The Expanded Disability Status Scale (EDSS) was used to quantify the severity of MS. Patient activities were monitored before and after standard immunomodulatory therapy (glatiramel acetate or interferon beta), and autonomy was observed and quantified. 

### 2.3. Materials and Measures

#### 2.3.1. 15D-Instrument

The health-related quality of life (HRQoL) was assessed with the general instrument 15D that was validated in patients with chronic pain [30]. The 15D is a 15-dimensione self-administered instrument that describes mobility, vision, hearing, breathing, sleeping, eating, speech, excretion, usual activities, mental function, discomfort, symptoms of depression, distress, vitality, and sexual activity [31]. In the current study, the Romanian language version of the 15D was used [32]. The generated 15D score is a single-index number on a 0–1 scale, where 0 indicates deceased and 1 refers to perfect health.

#### 2.3.2. Personal Autonomy Questionnaire (PAQ)

PAQ is a Romanian 36-item questionnaire designed to assess four dimensions of PA: cognitive (9 items), behavioral (11 items), emotional (8 items), and value autonomy (8 items), where higher scores reflect a greater personal autonomy [33]. 

The resulting scores for every dimension and the total value of PA, named T quotas, are considered “high” if they are higher than 60 and “small” if they are lower than 40. The range of T quotas scores is 0–100. 

If, for example, the T quotas for cognitive autonomy is 65, the patient has the ability to make decisions alone, critically analyzes the received information, forming his/her opinions without being influenced by others, and can self-evaluate. If, for example, the T quotas for behavioral autonomy is 35, the patient acts as others dictate to him or as he thinks others would, needs encouragement during his/her actions, and gives up performing difficult tasks if not helped. Small levels of emotional autonomy mean the patient avoids expressing his/her feelings when they are different from those of others or when he/she does not know what others are experiencing. A high level of value autonomy means patients do not give up their own beliefs and principles just because they are different from those of others or because those around them do not agree with them.

### 2.4. Statistical Analysis

All statistical analysis was performed with SPSS version 20. Descriptive statistics were used to map patients’ characteristics using percentages for categorical variables and mean ± standard deviation (SD) for continuous variables. The two groups were compared using Fisher’s exact test for categorical variables and Mann–Whitney U test for continuous variables; *p*-values less than 0.05 were considered statistically significant.

## 3. Results

All 26 patients completely answered the questionnaires. There were 15 patients in the RRMS group and 11 patients in the SPMS group (Table 1). The mean age of the patients was 43 years (24–69 years), with SD = 10. The age in the SPMS group was higher (*p* = 0.032) than in the RRMS group. There were more women than men in our study (23 vs. 3), but the distribution of gender was the same across the two groups (*p* = 0.743, >0.05). We observed that a higher proportion of patients with RRMS were female compared to those with SPMS (50% with RRMS vs. 38% with SPMS), but the difference was not statistically significant. We observed also the same distribution between the two groups for marital status (*p* = 0.442), level of education (*p* = 0.977), and environment status (*p* = 0.509). More patients in the RRMS group were working (*p*-value for employment status = 0.014). The mean of disease severity for RRMS patients (2.87) was smaller than the mean of disease’s severity for SPMS patients (4.27), but not statistically significant (*p* = 0.061).

The mean duration of MS was 9.5 years (SD = 5.12), with no significant differences between the two groups. Table 2 reports the mean, standard deviations, and percentages of all variables related to the health status of the patients.

The mean QoL of MS patients was 0.66 (SD = 0.18) before a new treatment and 0.71 (SD = 0.16) after the new treatment, with no significant differences between the two groups. The same level of personal autonomy was described by both RRMS and SPMS patients (*p* = 0.357). 

### Changes in 15D Scores before and after a New Treatment

In the post-treatment group, we observed a statistical improvement of the QoL (*p* < 0.001). The differences between scores measured at the two times of treatment (before, T0, and after, T1) were statistically significant only for seeing (*p* = 0.039 < 0.05), breathing (*p* = 0.007 < 0.05), sleeping (*p* < 0.001), usual activities (*p* = 0.027 < 0.05), depression (*p* = 0.016 < 0.05), distress (*p* = 0.046 < 0.05), and vitality (*p* = 0.018 < 0.05). Table 3 reports the descriptive statistics for each of the health components and dimensions of the 15D instrument.

Figure 1 compares the mean values obtained for all 15 dimensions of the QoL instrument: mobility, vision, hearing, breathing, sleeping, eating, speech, excretion, usual activities, mental function, discomfort, depression, distress, vitality, and sexual activity. The largest differences were observed for the sleeping dimension.

The overall QoL was better after the new treatment than before its start, as shown in Figure 1 and Figure 2. The coefficient of determination, *R*^2^ = 0.972, was very high, meaning a higher predictable QoL for MS patients after a new treatment, with 97.2% of points for the QoL value falling within the regression line. After changing the treatment, the MS patients exhibited a clinically important higher mean 15D score than before the new treatment.

As shown in Figure 3, gender, marital status, employment status, environment, and duration of the disease were not significantly related to the QoL. Age and severity of disease were negatively correlated, whereas education was positively correlated with QoL. Further, no type of autonomy was significantly related to the QoL. Only value autonomy was positive correlated (Pearson coefficient = 0.397, *p* < 0.05) with usual activities (as: employment, studying, housework, free-time activities) assessed for QoL.

In the heatmap, the red color was assigned to the lowest value and the green color to the highest value of the Pearson’s coefficient. The variables were: 1 (Age), 2 (Gender), 3 (Marital status), 4 (Employment status), 5 (Education), 6 (Environment), 7 (Duration of MS), 8 (Severity of MS), 9 (QoL before the new treatment), 10 (QoL after the new treatment), 11 (Cognitive autonomy), 12 (Behavioral autonomy), 13 (Emotional autonomy), 14 (Value autonomy), 15 (Autonomy), T1 (Time of measurement after the new treatment), 16 (Moving_T1), 17 (Seeing_T1), 18 (Hearing_T1), 19 (Breathing_T1), 20 (Sleeping_T1), 21 (Eating_T1), 22 (Speech_T1), 23 (Excretion_T1), 24 (Usual Activities_T1), 25 (Mental_T1), 26 (Discomfort_T1), 27 (Depression_T1), 28 (Distress_T1), 29 (Vitality_T1), 30 (Sexual activity_T1).

## 4. Discussion

RRMS is characterized by clinical attacks with complete or incomplete clinical recovery. Between the periods of disease recurrence, there is a minimal evolution of signs and symptoms, and recurrences can leave residual disabilities.

SPMS is characterized by partial recovery after attacks and continuous progression with or without occasional recurrences, minor remissions, and plateaus. The SP form represents the transformation of the RR form after approximatively 10 years. However, there are no criteria to determine when RRMS will turn into SPMS. In this study, Romanian patients with RRMS and SPMS reported the same QoL and personal autonomy. They had different demographic characteristics: patients with RRMS were younger and, most often, working. The same results regarding socio-demographic variables and perceived general health were described by Gross et al. for a representative sample [34]. Also, as can be seen from the Table 2, there was no significant difference between the two forms of MS regarding EDSS, disease duration (years), QoL, cognition, behavior, emotional state, autonomy. In addition, Table 3 shows that disease evolution before and after treatment was similar in the two groups of patients. The literature review showed a lack of research studies about differences between RRMS and SPMS patients with regard to quality of life or personal autonomy. Little is known about the quality of life and personal autonomy of MS patients in relation to disease subtype.

In the last decade, epidemiological studies established that MS susceptibility is significant higher for women than for men [35]. In addition, the disease evolution pattern showed that women are more affected by RRMS than men [35,36]. Our study groups confirmed this pattern, though a limitation was the small number of males available to complete the study. Previous studies focused on the mechanisms that underlie this pattern [35,36]. Since many factors can be involved in this gender discrepancy (e.g., environmental changes, educational level, employment status), one possible explanation is that women lifestyle changed significant in the past decades, and this can be an important factor affecting the quality of life. However, there is a large variation between EU countries, including Romania, regarding lifestyle factors, which have to be taken into account in MS treatment strategies in order to achieve better outcomes. These factors include the access to novel treatments information, cultural influences on daily life, and disease perception. Given that MS outcomes can be influenced not only by the molecular and cellular differences between people but also by social, economic, cultural, and environmental factors, these should be carefully assessed and considered when evaluating differences between EU countries. Our study is a pilot study aimed to identify a valuable tool that can be used in the assessment of the QoL in a particular cultural setting to improve health policy strategies.

Patient-reported outcomes (PROs) are very important in evaluating the effectiveness of MS treatments. Our study showed that QoL is not correlated with patient autonomy: some patients insisted on their autonomy, while others did not, but all patients reported an increase in their QoL. Franklin et al. proved that QoL for patients with MS is dependent on patients’ perceived ability to maintain autonomy and control [37]. Patients’ reasons for choosing between options are foremost unimportant as long as the QoL grows. According to the classification by Alanne et al. of the differences in the 15D scores [38], our study demonstrated an improvement with a much better change of 0.05 in the 15D score, even if not for all the dimensions assessed by the 15D instrument.

For example, a female patient of 33 years and medium autonomy (score = 46) had small cognitive autonomy (score = 36), indicating that she would need the help of others when making decisions, did not trust her abilities, let herself be influenced by the opinions of others. However, she was helped by friends and family, and her QoL was very high (value = 0.94). Alonso-Sardon et al. reported that patients with a higher degree of personal autonomy as a result of institutional and family support have a better health and QoL [39].

The QoL of MS patients proved to be dependent on age, education, and severity of the disease: a better QoL was reported by younger patients, more educated, and with less severe MS.

This study has several limitations. The statistical assessment of QoL in a relatively small population is difficult, as outliers can change the outcomes. In our study, we found no significant differences between groups for gender, but this can be due to the small number of available males. The trend we found should be further validated on a large cohorts of males in a multicentric clinical study. Thus, the results are consistent, but some other covariates could influence the QoL, such as medication adherence, pregnancy, or income. In order to quantify the HRQoL, we need a valid tool for the Romanian population, based on a precise and valid measurement algorithm for MS patients. There is still a lack of instruments for the assessment of HRQoL in MS patients in Romania, and this difficulty has to be solved in the future. In our study, we used two of the few valid available tools, i.e., the 15D-instrument and the PAQ, which allow a high number of measurements with a single instrument. However, these instruments can be further improved by including other specific MS measurement points (e.g., number of children or income bracket) already validated in our region and thus suitable to be used for MS patients. The results from the literature suggest that individuals who are more autonomously self-regulating tend to have a high treatment adherence [38]. Both indirect (self-reported, pill counts, electronic databases) and direct (measurement of drug/metabolite levels) methods for measuring treatment adherence can help researchers and clinicians. More PROs [40] or other tools should be applied to find factors that might be associated with a worse QoL. The degree to which patients perceive they have some control over events in their life and are not overly dependent upon others is related to a non-depressed condition. 

QoL is a complex appreciative element, which comprises at least three domains: physical, mental, and social. This concept deals with the impact of a disease and/or treatment on patients or their caregivers and with life satisfaction.

Most neurodegenerative diseases have, doubtlessly, an impact on the quality of life, and we have documented this in the current manuscript. For example, in Parkinson Disease (PD) patients, cognitive dysfunction occurs in 40–65% of the patients and depression in 40–60% of the them. Other factors that are related to poorer QoL are fatigue and motor disabilities, which occur in MS patients [41]. More than 70% of patients with Alzheimer’s Disease (AD) are dependent on their families, and their illness has a significant impact on their families. It has been shown that in patients with AD, cognitive function, apathy, irritability, and depression influence their QoL [42].

As the data show, QoL of patients with neurodegenerative diseases such as MS, PD, AD, and Amyotrophic Lateral Sclerosis (ALS) is less influenced by patient’s autonomy and more influenced by fatigue and state of mind. There is no common definition of QoL, and measurement scales can be generic or specific for a disease. The election of a QoL scale depends on the user’s aims.

The principal finding of this study is that after starting a new treatment that increases the QoL, the score for patient’s autonomy does not influence the QoL. At the same time, personal autonomy was only partially associated through value autonomy with better outcomes for usual activities like employment, studying, housework, free-time activities.

## 5. Conclusions

Value autonomy, which contribute to QoL, influences the usual activities of MS patients. The value of QoL is dependent on age, education, and MS severity. In contrast, personality traits (e.g., personal autonomy) could not be proved to be key factors influencing the QoL of patients with MS.

## Figures and Tables

**Figure 1 jcm-09-01349-f001:**
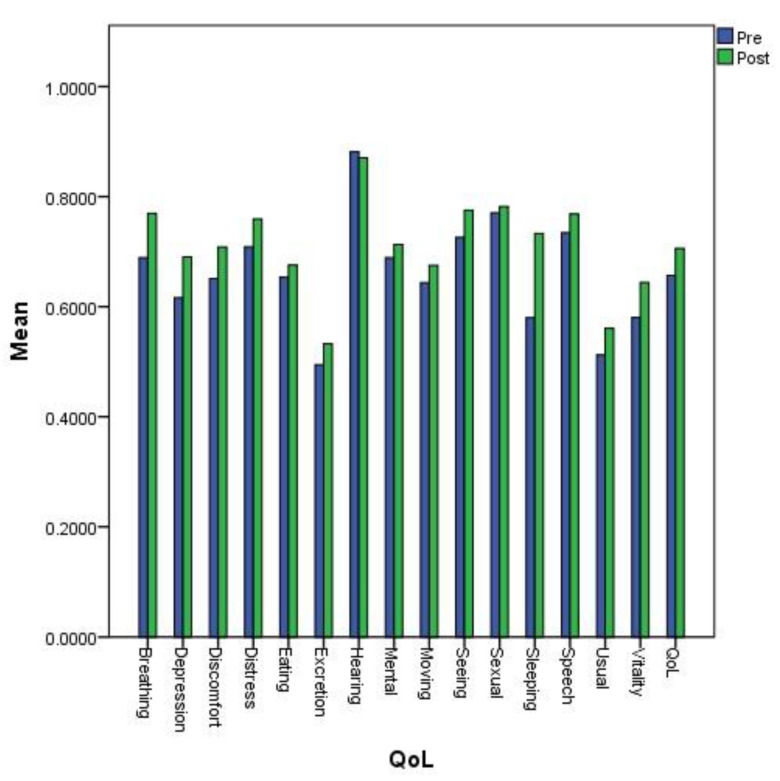
Mean values of the 15 dimensions determining Quality of Life (QoL) before and after a new treatment.

**Figure 2 jcm-09-01349-f002:**
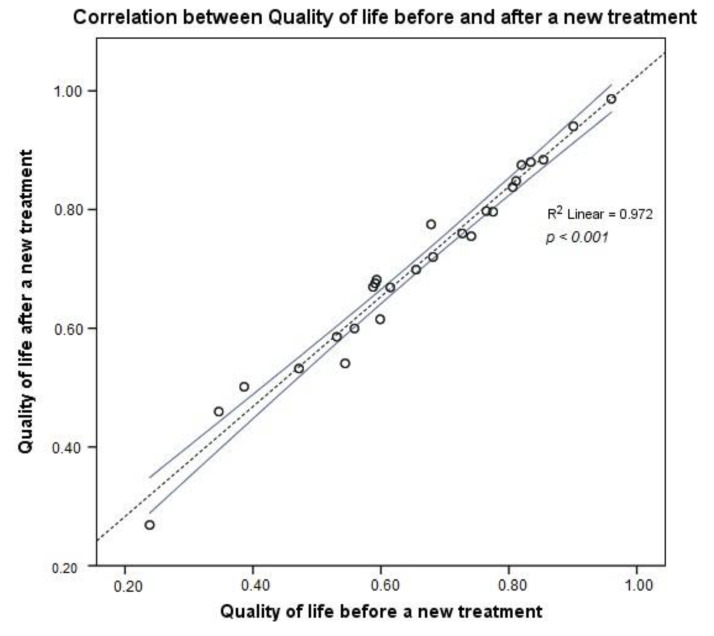
Correlation of QoL values before and after a new treatment.

**Figure 3 jcm-09-01349-f003:**
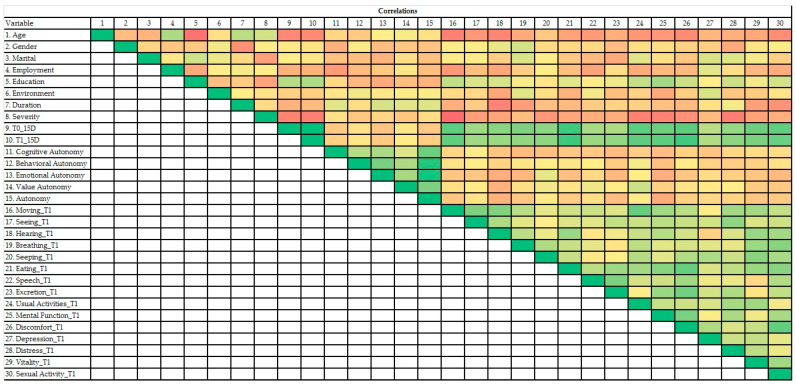
Heatmap of variables’ correlations found in this study.

**Table 1 jcm-09-01349-t001:** Demographics of the multiple sclerosis (MS) patients examined in this study.

Variables	Patient Group				*p*-Value
	Relapsing–Remitting MS (*N* = 15)		Secondary Progressive MS (*N* = 11)		
	Count	Mean (±SD)	Count	Mean (±SD)	
Age, years (43 ± 10)		39.7 (±5.97)		47.5 (±12.7)	0.032
Gender					0.743
Men (*n* = 3)	2 (8%)		1 (4%)		
Women (*n* = 23)	13 (50%)		10 (38%)		
Marital status					0.442
Married (*n* = 18)	11 (42%)		7 (27%)		
Unmarried (*n* = 6)	4 (4%)		2 (8%)		
Divorced (*n* = 1)	0		1 (4%)		
Widow (*n* = 1)	0		1 (4%)		
Employment status					0.014
Working (*n* = 3)	3 (12%)		0		
Not working (*n* = 17)	11 (42%)		6 (23%)		
Unemployed (*n* = 2)	0		2 (8%)		
Pensioner (*n* = 4)	1 (4%)		3 (12%)		
Environment status					0.509
Urban (*n* = 17)	9 (35%)		8 (31%)		
Rural (*n* = 9)	6 (23%)		3 (12%)		
Level of education					0.997
Middle school (*n* = 8)	4 (4%)		4 (4%)		
High school (*n* = 15)	10 (38%)		5 (19%)		
Faculty (*n* = 3)	1 (4%)		2 (8%)		

**Table 2 jcm-09-01349-t002:** Health status of the MS patients. EDSS, Expanded Disability Status Scale.

Variables	Patient Group				*p*-Value
	Relapsing–Remitting MS (*N* = 15)		Secondary Progressive MS (*N* = 11)		
	Count	Mean (±SD)	Count	Mean (±SD)	
**EDSS**	5 (19%)		1 (4%)		0.061
**1–1.5 (No disability) (*n* = 6)**	1 (4%)		2		
**2–2.5 (Disability is minimal) (*n* = 3)**	6 (23%)		0		
**3–3.5 (Disability is mild to moderate) (*n* = 6)**	1 (4%)		4 (4%)		
**4–4.5 (Disability is moderate) (*n* = 5)**	2 (8%)		1 (4%)		
**5–5.5 (Increasing limitation in ability to walk) (*n* = 3)**	0		2 (8%)		
**6–6.5 (Walking assistance is needed) (*n* = 2)**	0		1 (4%)		
**7–7.5 (Confined to wheelchair) (*n* = 1)**	0		0		
**8–8.5 (Confined to bed or chair) (*n* = 0)**	0		0		
**9–9.5 (Completely dependent) (*n* = 0)**	0		0		
**Duration of MS, years (9.5 ± 5.12)**		9 (±4.5)		10.18 (±6)	0.646
**Quality of Life_T0 (0.66 ± 0.18)**		0.69 (±0.15) Median = 0.68		0.61 (±0.2) Median = 0.60	0.330
**Quality of Life_T1 (0.71 ± 0.16)**		0.74 (±0.13) Median = 0.76		0.66 (±0.2) Median = 0.68	0.305
**Cognitive Autonomy**		39.53 (±10.44) Median = 41		36.18 (±8.7) Median = 35	0.721
**Behavior Autonomy**		51.4 (±14.77) Median = 51		45.1 (±13.7) Median = 37	0.217
**Emotional Autonomy**		45.9 (±7.7) Median = 44		45 (±5.5) Median = 46	0.281
**Value Autonomy**		43.53 (±10.27)		45.18 (±8.8)	0.683
**Autonomy**		44 (±12.23) Median = 44		40.54 (±9.3) Median = 39	0.357

**Table 3 jcm-09-01349-t003:** Changes in quality of life (QoL) for MS patients before and after a new treatment.

Dimensions	Pre-Treatment (Mean ± SD)	Post-Treatment (Mean ± SD)	Difference between Scores Post- and Pre-Treatment (Mean ± SD)	*p*-Value
**Moving**	0.64 (±0.25)	0.68 (±0.27)	0.03 (±0.09)	0.102
**Seeing**	0.73 (±0.21)	0.78 (±0.19)	0.05 (±0.10)	0.039
**Hearing**	0.88 (±0.17)	0.87 (±0.19)	−0.01 (±0.06)	0.317
**Breathing**	0.69 (±0.3)	0.77 (±0.25)	0.08 (±0.12)	0.007
**Sleeping**	0.58 (±0.21)	0.73 (±0.22)	0.15 (±0.13)	<0.001
**Eating**	0.65 (±0.26)	0.68 (±0.26)	0.02 (±0.08)	0.180
**Speech**	0.73 (±0.2)	0.77 (±0.22)	0.03 (±0.10)	0.083
**Excretion**	0.49 (±0.28)	0.53 (±0.27)	0.04 (±0.10)	0.059
**Usual Activities**	0.51 (±0.31)	0.56 (±0.3)	0.05 (±0.10)	0.027
**Mental Function**	0.69 (±0.25)	0.71 (±0.25)	0.02 (±0.09)	0.180
**Discomfort**	0.65 (±0.24)	0.71 (±0.26)	0.06 (±0.15)	0.165
**Depression**	0.62 (±0.2)	0.69 (±0.21)	0.07 (±0.15)	0.016
**Distress**	0.71 (±0.21)	0.76 (±0.22)	0.05 (±0.13)	0.046
**Vitality**	0.58 (±0.24)	0.64 (±0.19)	0.06 (±0.11)	0.018
**Sexual Activity**	0.77 (±0.22)	0.78 (±0.23)	0.01 (±0.06)	0.317
**Total**	0.66 (±0.18)	0.71 (±0.16)	0.05 (±0.03)	<0.001

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
