# Peer review of "Personal Autonomy as Quality of Life Predictor for Multiple Sclerosis Patients"

_jcm, 2020, doi:10.3390/jcm9051349_

Round 1

Reviewer 1 Report

The manuscript "Personal Autonomy as Quality of Life Predictor in Multiple Sclerosis" represents an original work carried out to draw a correlation between the QoL and personal autonomy. The work is interesting, however, it is required to correct the grammatical mistakes that could be spotted easily in the manuscript. In addition, I have a few suggestions that could improve the overall  quality of the manuscript

1. The authors have explained the correlation between QOL and personal autonomy, a short description of pharmacological treatment administered to the patients that could ultimately affect the QOL would help in determining the conclusion more efficiently.  

2. Reference number 37 should be looked into regarding its appropriateness to the study. 

3. Please use high-resolution Figures.

Author Response

Thank you for your time to review and comment. We have considered all your comments and found these useful in improving our manuscript.

Reviewer 2 Report

In the current study, Padureanu et al. evaluated relationships between quality of life (QoL) and personal autonomy using various patient measures in a cohort of 26 patients. Patient activities were monitored before and after treatment, and autonomy was observed and quantified. Some correlations were observed with respect to QoL, age and MS severity, with no correlation between QoL and patient autonomy. Although aspects of the study are interesting, some concerns may require further attention.

Comments:

Mostly females were used in the study (almost 90%). Some discussion with respect to gender differences in MS would be helpful, as well as gender-specific response to treatment, and quality of life (especially in relation to Romanian culture). Because of the small number of males in the study, it may not be wise to conclude that “no significant differences between groups for gender” in the abstract, especially given the small cohort sizes used in the analyses.

It was not clear whether any differences were seen in the RR and SP MS groups. Did the authors anticipate any differences between these 2 groups? Some clarification in the results descriptions or discussion may be helpful.

Some other details with respect to the patient selection criterion and other social measures would be useful to evaluate “quality of life”. Given the mean age, whether the patients would have children may also be a critical factor, as well as social status (income bracket).

Figure 2, please specify the exact p-value in the text. Also, detailed axis descriptions may be important (T01 and T1), as well as what the scales represent.

Figure 3 is currently very small, and somewhat difficult to interpret as a table. Can the authors reformat the table into a heatmap? A better description with respect to scale would also help. Also variable descriptions should be included in both axes in the table or heatmap, as readers currently need to look up numbered variables on the right in the small table print.

I also suggest that the authors consider summarizing positive and negative correlates from Table 3.

Currently, information with respect to “treatment” is not readily seen in the manuscript. Treatment details and duration would be helpful. Further, a discussion of patient outcome with similar or identical treatment regimens may be of importance.

It may also be important to compare in the discussion, current findings in the study with other available similar studies with respect to QoL/autonomy in other neurodegenerative disorders.

I recommend a thorough proofreading. The paragraphs tend to be short, and descriptions could be improved as presented in the current version.

Author Response

(The authors gave the same response as above.)
